# Prevalence of Antibodies to Japanese Encephalitis Virus and Severe Fever with Thrombocytopenia Syndrome Virus in Wild Boars Captured Across Different Locations in Toyama Prefecture, Japan

**DOI:** 10.3390/v17121585

**Published:** 2025-12-05

**Authors:** Shunsuke Yazawa, Kotoha Yoshida, Kotaro Fujii, Yumiko Saga, Sara Taniguchi, Ryosuke Suzuki, Chang-Kweng Lim, Miki Ishida, Kazunori Oishi, Hideki Tani

**Affiliations:** 1Department of Virology, Toyama Institute of Health, Toyama 939-0363, Japan; shunsuke.yazawa@pref.toyama.lg.jp (S.Y.);; 2Eastern Toyama Prefectural Livestock Hygiene Service Center, Toyama 939-3536, Japan; 3Department of Virology II, National Institute of Infectious Diseases, Japan Institute for Health Security, Tokyo 208-0011, Japan; 4Department of Virology I, National Institute of Infectious Diseases, Japan Institute for Health Security, Tokyo 162-8640, Japan; 5Toyama Institute of Health, Toyama 939-0363, Japan

**Keywords:** wild boars, JEV, SFTSV, SRIPs, antibody

## Abstract

Wild boars inhabit fields, hills, and farms across Japan, where they are fed on by numerous arthropods, including mosquitoes and ticks. Consequently, they are frequently exposed to arthropod-borne pathogens. In Toyama Prefecture, blood samples from captured wild boars have long been collected for classical swine fever virus antibody testing, with detailed records kept on the capture locations. In this study, we investigated the prevalence of antibodies against Japanese encephalitis virus (JEV) and severe fever with thrombocytopenia syndrome virus (SFTSV) using 3059 serum samples collected from wild boars over the past six years. A previously developed single-round infectious particles (SRIPs) assay system was employed for the analysis. We also examined the geographic distribution of antibody-positive wild boars. The results showed that antibody positivity rates for both JEV and SFTSV increased annually from 2019 to 2024. Geographical analysis revealed that JEV antibody-positive wild boars were distributed throughout Toyama Prefecture, whereas SFTSV antibody-positive wild boars were concentrated mainly in the northwestern region and along the western prefectural border. These findings suggest that JEV continue to pose an infection risk across the entire prefecture, while SFTSV has been actively spreading in the northwestern area during 2023–2024, raising concern over an increasing risk of human infection.

## 1. Introduction

In recent years, humans have come into closer contacts with natural environments such as mountains and rural landscapes, thereby increasing the risk of exposure to zoonotic pathogens. As a result, the likelihood of zoonotic infections in humans has grown, either through direct contact with wildlife or via arthropod vectors such as ticks and mosquitoes inhabiting these areas. Our institute has previously examined the prevalence of zoonotic infections in wild animals. However, these studies faced challenges in obtaining large sample sizes, and the uneven geographic distribution of survey areas limited the scope of their evaluation.

Since the confirmation of pigs infected with classical swine fever virus in Aichi Prefecture, Japan, in September 2018 [1], wild boars carrying the virus have been detected in multiple regions, promoting nationwide surveillance of classical swine fever in wild boars that continues to this day. In Toyama Prefecture, wild boars captured not only for classical swine fever monitoring but also for food safety and game meat inspection have provided an opportunity to obtain a substantial number of samples from across the prefecture.

Japanese encephalitis is a zoonotic disease caused by Japanese encephalitis virus (JEV), transmitted by *Culex tritaeniorhynchus* mosquitoes, and capable of causing encephalitis in humans [2]. Owing to routine vaccination programs in Japan, high antibody prevalence rates have been maintained in humans, and only a small number of human cases are reported annually [3]. Nevertheless, once the disease develops, it is associated with high mortality and severe sequelae, underscoring the need for vigilance from a public health perspective. Pigs and wild boars act as amplifying hosts for JEV [4,5], and antibody positivity rate in these animals provides a useful indicator of the local presence and prevalence of the JEV. In Japan, antibody surveys are conducted on unvaccinated fattening pigs less than six months of age. High antibody prevalence rates are reported annually, particularly in western Japan, confirming the continued circulation of JEV. We previously developed a method for antibody titer measurement using JEV single-round infectious particles (SRIPs), which is simpler than conventional assays [6,7]. The SRIP-based assay offers two key advantages: safety, since SRIPs are nonpathogenic, and simplicity, as infectivity is measured via luciferase activity and can be quantified mechanically.

Severe fever with thrombocytopenia syndrome (SFTS) is a zoonotic disease caused by the recently identified SFTS virus (SFTSV). This virus belongs to the family *Phenuiviridae* and is transmitted by ticks. It has caused serious illness in humans and companion animals such as dogs and cats. Antibody surveys have confirmed infection in wild animals, but the natural host has not yet been identified. It presents with fever and gastrointestinal symptoms and has a mortality rate of 10–30% [8]. Since its first detection in Japan in 2012, SFTS cases have been reported mainly in western Japan, but the disease has gradually expanded eastward, necessitating close monitoring. To facilitate SFTSV antibody detection, we developed a novel SRIP system based on a previously constructed SFTSV replicon [9]. As with JEV-SRIPs, luciferase activity was used as an indicator, allowing antibody titers to be measured easily. By employing artificial pseudo-virions incorporating a reporter gene, antibody titers can be assessed efficiently across large sample sets.

Accordingly, we examined the prevalence of antibodies against JEV and SFTSV in approximately 3000 wild boar serum samples collected and stored in Toyama Prefecture between 2019 and 2024 for classical swine fever surveillance. We then evaluated the antibody prevalence of JEV and SFTSV in the prefecture.

## 2. Materials and Methods

### 2.1. Wild Boar Serum Samples

This study analyzed 3059 serum samples obtained from wild boars captured in Toyama Prefecture between July 2019 and October 2024. The wild boars were randomly captured using traps and hunting guns, (Request for cooperation in conducting wildlife infection confirmation tests following the outbreak of swine fever in Gifu Prefecture, No. 1809146, 14 September 2018; Policy to strengthen the capture of wild boars to prevent the spread of swine fever, No. 6500-2, No. 3690-2, 1 April 2021), and were generally healthy living creatures. Age class was determined based on external characteristics. Individuals with juvenile coat pattern or small body size were classified as juveniles, whereas all other animals were classified as adults. Blood samples was collected directly from major vessels during field dressing. Samples were placed into serum-separator tubes and allowed to clot at ambient temperature before centrifugation. Serum was aliquoted, labeled, and stored at −80 °C until serological testing. These samples had been originally collected and stored for classical swine fever testing. All samples tested negative for classical swine fever and were subsequently inactivated at 56 °C for 30 min to prepare the test serum. Consent for their use in this research was granted by the local governments and organizations responsible for capturing wild boars.

### 2.2. JEV Survey

JEV-SRIPs were generated from three plasmids (pCMV-JEV-Beijing-1-prME, pCAG-YF-C, and pCMV-YF-nluc-rep), as previously described [10]. These plasmids were transfected into 293T cells (obtained from the JCRB Cell Bank), which had been seeded to confluence the previous day, using polyethyleneimine (Thermo Fisher Scientific Inc., Waltham, MA, USA). After incubation at 37 °C for 3 days, the culture supernatant was collected and used as JEV-SRIPs. Infectivity of JEV-SRIPs was quantified by luciferase activity. Final JEV-SRIPs dilution was adjusted to yield approximately 1 × 10^6^–10^7^ relative light units (RLU) in JEV-SRIPs only control wells. Then, mixed with test sera (final serum dilution: 1:100) for neutralization. A 40 µL aliquot of the reaction mixture was inoculated onto Vero Osaka cells (obtained from the National Institute of Infectious Diseases, 1-23-1 Toyama, Shinjuku-ku, Tokyo 162-8640, Japan), which had been seeded the previous day in a 96-well white plate (SPL Life Sciences, Daejeon, Republic of Korea), and incubated at 37 °C for 1 day. After incubation, the medium was removed, the cells were washed once with PBS, and 40 µL of luciferase substrate solution (Nano-Glo Luciferase Assay, Promega, Madison, WI, USA) was added. Luciferase activity was measured using a GloMax^®^ Navigator microplate luminometer (#G2000, Promega Corporation). Luminescence was recorded with an integration time of 1.0 s per well. Signals were expressed as relative light units. The infection rate was determined by dividing the measured value by the JEV-SRIPs control value, with samples showing an infection rate below 50% classified as antibody-positive. Low titer reactions in wildlife sera are known to contain nonspecific inhibition. To avoid false-positive interpretation, we adopted a conservative cutoff of 1:100.

In addition, antibody titers were calculated for antibody-positive samples. Sera were adjusted to final dilution ratios ranging from 1:100 to 1:800, and the infection rate was calculated as described above. A regression line was constructed using the infection rates at each dilution, and the 50% inhibitory concentration (IC_50_) was determined, which was taken as the antibody titer. In this study, IC_50_ values were used as relative indicators of neutralizing activity, not as absolute antibody titers, because the assay linearity is confined to the approximate range of 100–800. Accordingly, IC50 values < 100 were regarded as below the quantifiable range.

The performance characteristics of the SRIP-based neutralization assay have been previously evaluated. In our earlier studies, the assay showed strong correlation with conventional virus neutralization tests and established serological methods, supporting its analytical reliability [7].

### 2.3. SFTSV Survey

SFTSV-SRIPs were generated using four plasmids: pCAG-SFTSV(SPL-030)-GP, pKs-SFTSV(SPL-005)-L, pPdIV-SFTSV(SPL-005)-Mseg·EGFP-HiBiT, and pKs-SFTSV(SPL-005)-NP [9]. The procedure for producing SFTSV-SRIPs was identical to that used for JEV-SRIPs described above. In addition, infection by SFTSV-SRIPs was confirmed to be inhibited by serum from an SFTSV-infected patient. A schematic diagram of the measurement method is presented in Figure 1A. Assessment of luciferase activity enabled the processing of large sample numbers and allowed evaluation of neutralizing activity even with small viral quantities. Infectivity of SFTSV-SRIPs was quantified by luciferase activity. Final SFTSV-SRIPs dilution was adjusted to yield approximately 1 × 10^6^–10^7^ RLU in SFTSV-SRIPs only control wells. Then, mixed with test sera (final serum dilution: 1:100) for neutralization. A 40 µL aliquot of the reaction mixture was inoculated onto Vero Osaka cells, which had been seeded the previous day in a 96-well white plate, and incubated at 37 °C for 1 day. After incubation, the medium was removed, the cells were washed once with PBS, and 40 µL of luciferase substrate solution (Nano-Glo HiBiT Lytic Detection System, Promega, WI, USA) was added. Luciferase activity was measured using a GloMax^®^ Navigator microplate luminometer. Luminescence was recorded with an integration time of 1.0 s per well. Signals were expressed as relative light units. Antibody presence or absence was determined using the same method as for JEV-SRIPs. For samples that tested positive for SFTSV antibodies, IC_50_ values were calculated and defined as antibody titers.

### 2.4. Geographical Distribution Mapping

A geographical distribution map was created using the coordinate data of captured wild boars. QGIS 3.34 (QGIS Development Team, 2025) was used to visualize the coordinate data. The background map consisted of Carto Light (©CARTO, ©OpenStreetMap contributors, CC BY 3.0) and the National Land Numerical Information Administrative Area Data (Ministry of Land, Infrastructure, Transport and Tourism, Land Policy Bureau, https://nlftp.mlit.go.jp/ksj/gml/datalist/KsjTmplt-N03-2025.html?utm_source=chatgpt.com (accessed on 2 December 2025)). The locations of antibody-positive and antibody-negative wild boars were then plotted on the map.

## 3. Results

### 3.1. Wild Boar Serum Samples

The details of the wild boar serum samples are summarized in Table 1. Approximately one-third of the samples were from juvenile wild boars, while the remaining two-thirds were from adults. Since 2023, the capture numbers of both juveniles and adults have increased two-fold. The survey area is shown in Figure 1B. Toyama Prefecture, located in north-central Japan and surrounded on three sides by mountains, is believed to harbor a large wild boar population. The serum samples analyzed in this study represent about 20% of the total number of wild boars captured annually.

### 3.2. Annual Percentage and Distribution of JEV Antibody-Positive Wild Boars

Figure 1C was included to verify the quality and infectivity of the SRIPs used in this study. Demonstrating JEV-SRIPs exhibited approximately 10^4^-fold higher luciferase activity than background non-enveloped particles confirms that the assay was performed under appropriate conditions and that the neutralization results are reliable. This has already been confirmed in previous reports [7].

Among the 3059 serum samples tested, 426 were positive for JEV antibodies, yielding a positivity rate of 13.9%. The yearly positivity rates from 2019 to 2024 were 4.2, 4.4, 11.1, 10.8, 10.3, and 23.1%, respectively, showing an overall upward trend (Figure 2A). The 2024 positivity rate doubled compared with that of 2023, reaching the highest level recorded. Geographical analysis showed that JEV-seropositive wild boars were distributed across the entirety of sampling area (Figure 3A). The annual distribution is presented in Appendix A. When the prefecture was divided into eastern and western regions along the 137.1° east longitude, a commonly used boundary in Toyama Prefecture, no significant differences in antibody positivity rates were observed either annually or overall. Antibody titers were also analyzed by year (Table 2A). In 2024, coinciding with the sharp increase in antibody positivity, the proportion of serum samples with titers of 500 or higher in 2024 tended to be higher than in 2023.

### 3.3. Annual Percentage and Distribution of SFTSV Antibody-Positive Wild Boars

The infectious titer of the SFTSV-SRIPs prepared in this study were >10^3^-fold higher than those of background nonenveloped particles (Figure 1C). The assay was performed under appropriate conditions and the neutralization results were reliable as JEV-SRIPs.

Of the 3059 serum samples, 76 tested positive for SFTSV antibodies, corresponding to a positivity rate of 2.5%. The annual positivity rates from 2019 to 2024 were 0.6, 0.7, 0.0, 0.7, 1.9, and 5.0%, respectively, reflecting a marked rise during the past two years (Figure 2B). Geographical analysis revealed that SFTSV-seropositive wild boars were mainly detected in the northwestern and western regions of Toyama Prefectures, near borders with neighboring prefectures (Figure 3B). Annual distribution data showed that until 2023, positive wild boars were found exclusively in the western region; however, several positives were also detected in the eastern region in 2024 (Appendix A). Regional comparisons showed significantly higher positivity rates in the western region than in the eastern region. Furthermore, the proportion of serum samples with antibody titers ranging from 500 to 2000 increased in both 2023 and 2024 (Table 2B).

## 4. Discussion

In this study, the positivity rate of JEV antibodies in wild boars increased annually, doubling in 2024 compared with the previous year. This result is consistent with a survey of fattening pigs in a neighboring prefecture, where JEV antibody positivity rate also rose markedly in 2024 [11]. Separately from this recent trend, previous studies have shown that wild boars generally exhibit high exposure to JEV in Japan. For example, a survey in Wakayama Prefecture reported 83% seropositivity in 2009 [12]. In addition, 10 of the 25 municipalities participating in the national endemic surveillance program reported JEV antibody positivity rates of 80% or higher in fattening pigs [12]. In the present survey, the positivity rate of JEV antibodies in wild boars from Toyama Prefecture was not significantly higher than in earlier surveys conducted in surrounding regions. However, as JEV-seropositive wild boars were detected throughout the prefecture and their numbers continue to rise annually, the risk of JEV transmission has been increasing in recent years.

The positivity rate of SFTSV antibodies in wild boars was nearly undetectable prior to 2022 but has risen since 2023, doubling in 2024 relative to the previous year. A survey in western Japan, where SFTS is endemic, found SFTSV antibody positivity rates of 12% in wild boars from Oita Prefecture and 53% in wild boars from Kagoshima Prefecture [13,14]. Since the first confirmed human case of SFTS in Japan in 2013, most cases have been concentrated in western Japan [15]. More recently, however, cases have also been reported in central and eastern regions. Toyama Prefecture, located in the western edge of the eastern region, confirmed its first human case in November 2022 and a second case in June 2025 [16,17]. Two canine cases were also reported in 2022 [18], and a feline case was identified in August 2025 [19]. The increase in SFTSV antibody positivity among wild boars in this survey coincided with the emergence of human cases in Toyama Prefecture. Although current wild boar positivity rates remain lower than those in western Japan, these findings suggest an increasing risk of SFTSV infection in Toyama.

In this study, coordinate data from wild boar blood sampling sites were recorded when samples were stored, allowing mapping of seropositive and seronegative animals for both JEV and SFTSV by map. The results showed that JEV-seropositive wild boars were distributed throughout Toyama Prefecture, whereas SFTSV-seropositive wild boars were concentrated in the northwestern and western region, along border with neighboring prefectures.

*Culex tritaeniorhynchus* mosquitoes, the vector of JEV, can travel several hundred meters per day, enabling widespread transmission [20]. By contrast, ticks typically move only a few meters daily [21]. Although ticks can be transported longer distances by host animals, their lower transmission efficiency compared with mosquitoes may explain the observed distribution patterns. Notably, in 2024, several SFTSV-seropositive wild boars were identified in the eastern part of Toyama Prefecture. A previous study on raccoons in Wakayama Prefecture also demonstrated a year-by-year increase in SFTSV antibody positivity, suggesting that the virus may spread across regions within a short time [22]. These results raise concerns that SFTS could rapidly expand throughout Toyama Prefecture in the near future.

Antibody titers were also measured in positive samples. For JEV, the proportion of sera with titers of 500 or higher increased by >20% in 2024 compared with the previous year, coinciding with the sharp rise in new infections. Similarly, for SFTSV, the proportion of samples with titers of 500 or higher increased between 2023 and 2024, in parallel with the rise in positivity rates. For both viruses, the antibody titer categories used in this study may provide a means of screening for recent infections.

As a new approach, we employed SRIPs to evaluate large numbers of wild boar serum samples for JEV and SFTSV antibodies. SRIPs are artificial pseudo-virions that are nonpathogenic, ensuring operator safety and ease of use. Infectivity can be quantified by luciferase activity, allowing rapid, objective, and automated assessment [7]. In this study, over 3000 samples were tested for JEV and SFTSV antibodies, and positive samples were subsequently titrated at four dilution points. The entire process was completed within three months. In contrast, conventional methods such as the focus method, TCID_50_ assay, would likely have required more than a year. Thus, SRIP-based antibody assays are straightforward and well suited for large-scale sample processing.

This study also benefited from the availability of serum samples collected for nationwide classical swine fever testing in Japan. These surveys are conducted not only to monitor classical swine fever but to ensure food safety in wild game and to prevent the spread of infected wild boars. Consequently, many wild boars captured in the prefecture are subject to testing. By using these existing samples, we were able to obtain a large dataset without additional effort.

Previous studies have reported lower SFTSV antibody positivity in wild boars than in deer and raccoon dogs [13], likely due to fewer ticks attaching to wild boars [23]. Therefore, the true abundance and distribution of SFTSV-infected ticks may be greater than suggested by wild boar antibody positivity rates in this survey.

Wild boars live year-round in forested environments and are continuously exposed to mosquito and tick bites without any behavioral or physical protection. Therefore, their antibody positivity reflects the intensity of virus circulation among arthropod vectors in the area. By applying SRIP-based methods to investigate antibody prevalence in a large number of wild animals and mapping the results, this study provided a clearer understanding of the risk of JEV and SFTSV transmission to humans in the study area. In Toyama Prefecture, JEV remains a public health risk, while SFTSV is currently concentrated in the northwestern region but appears to be spreading, indicating a rising risk of SFTSV infection. Both JEV and SFTSV infections in swine and wild boars are generally subclinical [24,25], and although domestic human JEV cases are rare and routine mosquito surveillance data for JEV in Toyama are limited, SFTS cases have been increasing nationwide in recent years, underscoring the importance of continued vector and human surveillance to better interpret these findings.

## Figures and Tables

**Figure 1 viruses-17-01585-f001:**
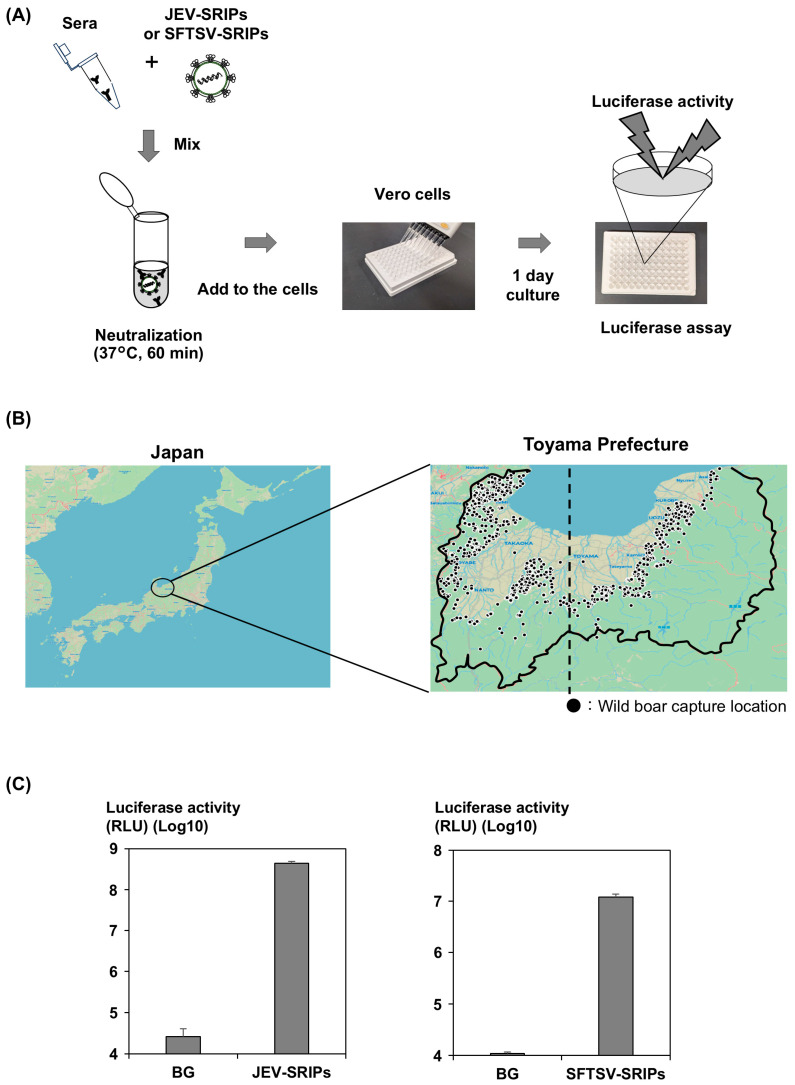
Survey information. (**A**) Schematic representation of the SRIP assay method. Wild boar serum was mixed with JEV-SRIPs or SFTSV-SRIPs and neutralized at 37 °C for 1 h. The reaction mixture was then inoculated into Vero Osaka cells, and luciferase activity was measured after one day of incubation. (**B**) Location of Toyama Prefecture. Toyama Prefecture is situated near the center of Japan, facing the Sea of Japan, and covers an area of 4247 km^2^, approximately 70% of which consists of forest and mountainous land. The solid line represents the Toyama Prefecture border. The dashed line represents the statistical 137.1° east longitude, which is the boundary between the eastern and western parts of the prefecture. (**C**) Infectivity of JEV-SRIPs and SFTSV-SRIPs. JEV-SRIPs or SFTSV-SRIPs exhibit approximately 10^4^- or 10^3^-fold higher luciferase activity than the background (BG), respectively. Error bars represent standard deviation (SD) from three independent experiments.

**Figure 2 viruses-17-01585-f002:**
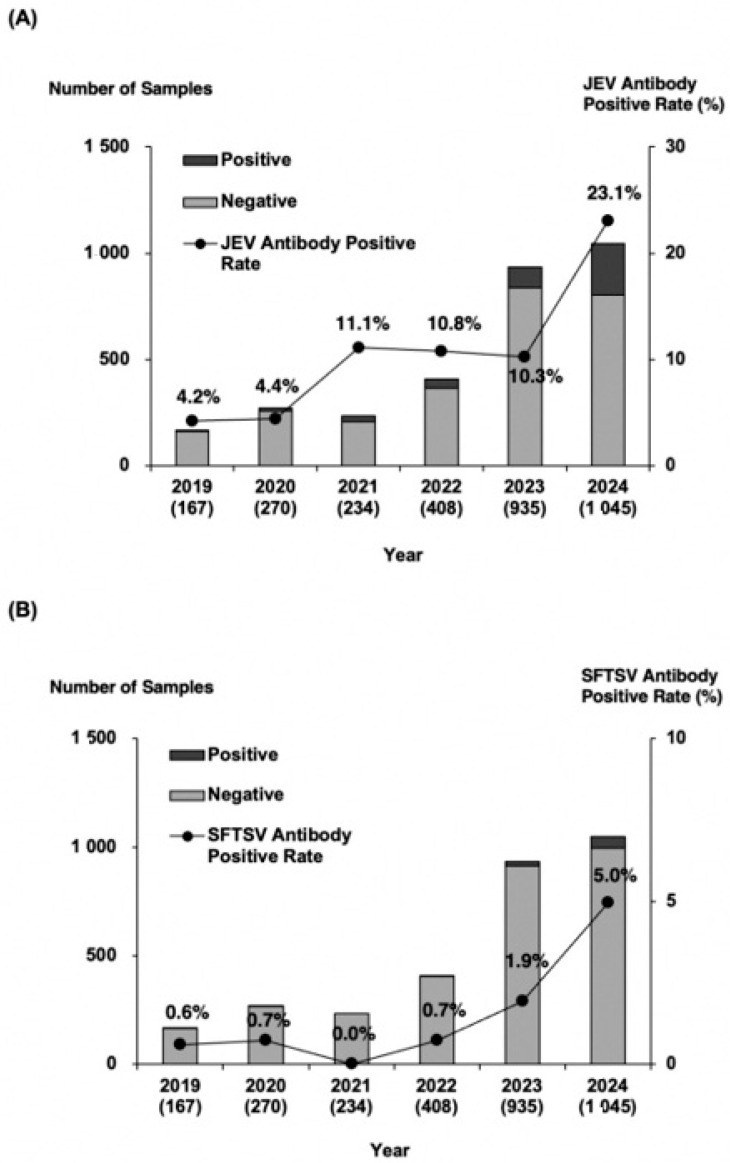
Annual number of wild boar serum samples and antibody positivity rates. (**A**) JEV survey and (**B**) SFTSV survey. The bar graph shows the total number of wild boars, with black bars indicating antibody-positive samples, and gray bars representing antibody-negative samples. The numbers in parentheses represent the total samples collected each year. The line graph illustrates the annual trend in antibody positivity rate, demonstrating a consistent increase over time.

**Figure 3 viruses-17-01585-f003:**
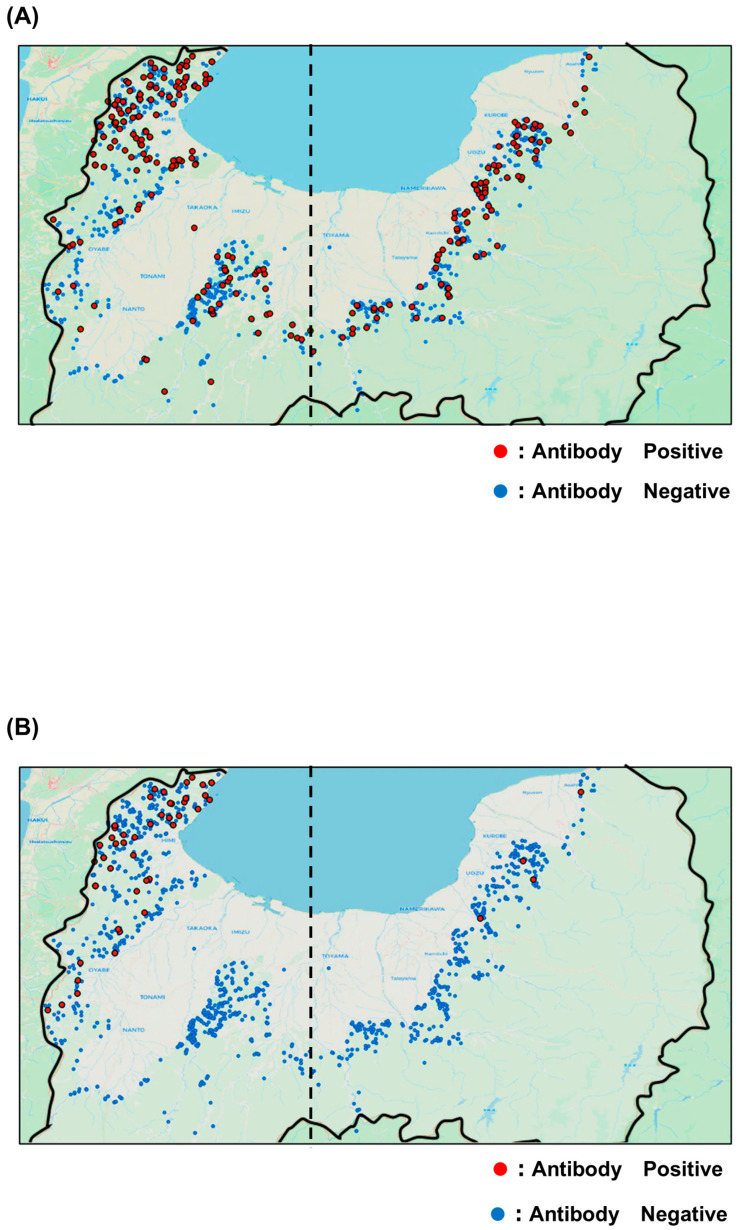
Geographic distribution maps. (**A**) JEV survey and (**B**) SFTSV survey. Red dots indicate capture locations of seropositive wild boars, while blue plots indicate capture locations of seronegative wild boars. The dashed line represents the statistical 137.1 °C, which is the boundary between the eastern and western parts of the prefecture.

**Table 1 viruses-17-01585-t001:** Details of wild boar serum samples.

		Juvenile	Adult	Unknown	Total
Year	2019	49	117	1	167
	2020	48	222	0	270
	2021	48	186	0	234
	2022	117	290	0	407
	2023	290	645	0	935
	2024	358	688	0	1046
Total		910	2148	1	3059

The table presents the number of wild boar serum samples collected each year from 2019 to 2024. Juvenile wild boars are defined as animals younger than 6 months of age. The number of samples has increased as the government is encouraged to test wild boars captured for meat purposes from mid-2022.

**Table 2 viruses-17-01585-t002:** Distribution of antibody titers in antibody-positive wild boars.

**(A)**				
		**JEV-Antibody Titer (IC_50_)**		
		**100–500 (%)**	**500–2000 (%)**	**>2000 (%)**
Year	2019	71.4	14.3	14.3
	2020	41.7	33.3	25.0
	2021	42.3	30.8	26.9
	2022	54.5	43.2	2.3
	2023	54.2	35.4	10.4
	2024	33.6	48.1	18.3
Total		41.8	42.7	15.5
**(B)**				
		**SFTSV-Antibody Titer (IC_50_)**		
		**100–500 (%)**	**500–2000 (%)**	**>2000 (%)**
Year	2019	0.0	100.0	0.0
	2020	0.0	100.0	0.0
	2021	−	−	−
	2022	100.0	0.0	0.0
	2023	27.8	61.1	11.1
	2024	38.5	55.8	5.8
Total		36.8	56.6	6.6

(A) JEV survey and (B) SFTSV survey. IC_50_ values were grouped into three categories (100–500, 500–2000, >2000).

## Data Availability

All original data are included in this article and the Appendix A. Further inquiries may be directed to the corresponding author.

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
