# Peer review of "Prevalence of Antibodies to Japanese Encephalitis Virus and Severe Fever with Thrombocytopenia Syndrome Virus in Wild Boars Captured Across Different Locations in Toyama Prefecture, Japan"

_viruses, 2025, doi:10.3390/v17121585_

Round 1
Reviewer 1 Report
Comments and Suggestions for Authors
The paper by Yazawa and colleagues is the report of a serological investigation for extimating the prevalence of two arbo-viruses on wild animals in a region of Japan.
The introduction is brief but sufficiently explanatory, except for minor points (see below).
The methods need more clarifications,as noted in the minor points.
The results section also requires some modification, as noted below in the minor points, which I kindly ask the authors to take into full consideration. In particular, the point which probably would require a better presentation is the graphic representation of Ab titres.
I think that the presented table 2 is not the best presenting option, while a figure with the actual distribution of sample numerosity vs measured Ab titres is highly desirable and maybe necessary.
The same figure could represent both cumulative and yearly distributions, and could also give representation of possible statistically singnificant differences.
Another point (also present in the minor points) that should better be explained in the section (but maybe also in the methods) is about the sampling. It would be interesting
to understand if sampling is somehow driven by some external factors, some specific strategy, or maybe some animal behaviour, and how these elements can have been varying in time.
This is aimed to exclude (or, in case, report) possible bias that could produce higher prevalence rates than in total population, or biased differences in prevalence between years.
So I think some explanatory delucidations could be provided in the text.
Furthermore, I would like to ask the authors if it possible for them to provide an investigation, even in a smaller scale, for viral RNA in the sampled sera. Are there any molecular
evidences of JEV or SFTSV in wildboars or other wild animals in the area in those years? Also, is there any possibility that some of the captured animals were bearing symptoms for JE or SFTS?
Last, since the authors have records of the animals age, they could try to see if there's a significant difference in terms of Ab titres between younger and older animals.
The discussion would benefit a lot from a figure depicting Japan prefectures. Also, it would be interesting to discuss the possibility that viruses are geographically spread not only by vectors, but also by hosts, by
adding, if available, information about animal moving routes across the region.
The declared focus of the study is on the epidemiological results rather than on methods, but since the authors add some discussion about the efficacy of the method, I would like them to also add some reminders about sensitivity-specificity of the method. This are not declared in the methods section, and maybe have been determined
in prior studies. In this specific one they haven't, which is not so satisfactory. I don't think it's possible for the authors to run in parallel a subset of samples with an independent method, but this could be interesting at least to discuss about.
Summarizing, I suggest the authors to take some time to try to assess the points discussing above, including also those highlighted in the following minor points section, trying if possible to add more data or reorganize the present ones into a clearer version
A few minor points:
- line 13: throughout the abstract, ";" is used in place of ","
- line 18: please introduce SFTSV acronym
- line 19: please define SRIP acronym
- line 31: although these elements can of course have a role in the spread of zoonotic pathogens, it is an excessive simplification to think that they were the main
causes. Other causes are by far more impactant on the phenomenon at a global level.
- line 45: have provided
- line 53: please add a supporting reference (for pigs and wildboars as JEV hosts)
- line 63: please add information about SFTSV classification, its host range and vectors
- line 79: please state as the animals were selected for sampling randomly or if there could have been a preference for "sick-looking" individuals. It is an important point.
- line 86: as a general curiosity (that readers may have as well), is it so well established that chimeric virions have the same serum-neutralization properties of actual JEV particles?
- line 91: please add information about the used infectious titre, and the method used to determine it
- line 102: how was the 1:100 threshold for positivity established? Please clarify in the text. Isn'it possible that some sera could be Ab +, but with a titre below 1:100?
- line 106: Information provided about SFTSV SRIP assay in unsufficient. As far as I can understand by reading into the reference papers, this assay is not luciferase-based, but uses a differet
chemiluminescence source, and probably different reagents. In this case, please also correct the sentence at line 69. In any case, please clarify by providing more accurate description of this protocol.
- line 107: please add numeric reference
- line 118: please change the citation to numeric format reference
- line 122: SRIPs measurement method must be better explained in the methods section, not here in the results
- line 125: it is not clear from the text wether this increase in captures reflects an increase of the population, or is due to different sampling strategies, or to other reasons. Please clarifiy if possible
- although I'm not familiar with the geography of Japan, it looks like figure 1A doesn't show the entire map of the prefecture, with no distinguishable borders and unreadable city names,making it really difficult to identify the capture areas for non native readers. I suggest to depict a wider area,
with clearly visible prefecture borders, and higher resolution typed toponyms. Please, Apply this suggestion to all maps in the article and supplementary files.
- line 127: looking at the map in figure 1, the prefecture doesn't seem to be in northern, but only central part of the country.
- line 128: the fact that the analysed sera are only a fraction of the available samples was never mentioned before (i.e. in the methods section). So, please go back to the relevant section (par 2.1), and explain
how was this sub-cohort generated (randomly, or on the basis of which parameters). This is always important in a seroprevalence evaluation study.
- line 133: this entire period is of difficult comprehension and unclear scientific meaning. The last sentence could maybe be used in the discussion of the methods performance. The first is completely unclear to me, as is the
related figure 1B: the titre of produced SRIPs (which is not what is shown in figure B, by the way) is not relevant for the results, as they must be used in a defined titre for the test, so this specific text line and figure is of no use.
Finally, the second sentence, about unprovided data of human serum neutralization, is absolutely unrelevant. I suggest to erase the entire period.
- line 140: the choice of merging three completely indipendent elements in a single figure 1 seems to be inappropriate. In particular, the C section is clearly more suitable for the methods section, to which I suggest
it to be moved. Part A could become a separated fugure, part B is in my opinion unrelevant and unrequired, and could be eventually moved to the supplementary section.
- line 151: a positivity rate of 13,9%
- line 155: they're not spread across the entire prefecture, they're spread across the entire sampling area (or capture region).
- line 156: Supplementary Figure 1 shows that sampling area differed substantially overtime, with an large increase of captures in the NW area in recent years. Many of the sampled sera in this area are shown to be positive, contributing to the general increase trend of the prefecture. My question is:
why this modification of sampling area? is it due to a change in sampling strategy or to a change in animal distribution? Could this be explained in the text?
- line 158: to show, at least qualitatively, those data, it would be sufficient to draw the meridian line on the provided maps.
- line 163, 169: the current organization of fig 2 and 3 is not the most rational one. Assembling together fig 2A and 3A in a new figure 2, and Fig 2B and 3B in a new figure 3 would much better suit the text organization
- line 182: this is not similar to JEV survey
- line 183: in this case it would be even more relevant to show this data. It is surely possible to both represent them in the map with a longitude line, and to calculate them in a table.
- line 186: in this table, and in the main text, regression-generated IC50 are used as estimates of Ab titres. Which is acceptable, with the limitation of not exceeding the linearity area of the biological assay. So, in absence of clear indication of a specific linearity pre-testing,
the linearity of the assay can only be assessed within the text values range, which is 100-800. So, adding a values category of 500-2000 (by the way, on the basis of what was this range generated?) is not so reliable. A category of >2000 is completely not applicable, as it's too far away from the value range of the assay.
Sorry, but in the absence of specific assaing for these high-range titres, this data is not very sound, and may not be used for downstream evaluations or discussions.
- line 189: this is a table, not a figure, so no caption is expected below the table. The table is self-explanatory, although it contains some errors (i.e. excess % symbols in the headers).
- line 196: I cannot verify this information as there's no link to ref 4, but I think it's unlikely that a report from 2022 can contain data from 2024.
- line 197: this paper (ref 9) is in accordance with the claim that high JEV prevalence is expectable in wildboars in Japan, but not with the observation of a steep increase over time, as the 83% data comes from a 2009 paper. So please rephrase
- line 203: is it possible to support this hypothesis with any incidence data from wild animals, or human subjects?
- line 210,211: eastern japan near western border is a confusing sentence, please rephrase.
- line 212, 213: please add references for human and feline SFTSV detection respectively
- line 237: these observation would be much better supported by a molecular testing looking for acute infections
- line 269: please note that one of the authors is listed as only contributing resources, which is not a sufficient criteria for authorship, but for simple acknowledgement
- line 296: please add a url or a doi for publication retrieval
Author Response
Thank you very much for reviewing our manuscript. Please see the attachment.

Reviewer 2 Report
Comments and Suggestions for Authors
I found the report describing serosurveillance for JEV and SFTSV by Yazawa and colleagues interesting and a valuable contribution to this field. The manuscript is generally well written and clearly described. I do not have major criticisms but offer the following suggestions:
Title: .. Toyama Prefecture, Japan (I would never search for Toyama but would for Japan)
Line 15: suggest “In Toyama Prefecture, blood samples from captured wild boars”
Line 19: need to define SRIPs
Line 45: delete “been” from “been provided”
Line 50: antibody prevalence rates have been maintained [in humans]
Line 111: do you mean “For samples that tested positive for SFTSV” ?
Line 135: In addition, infection by SFTSV-SRIPs was confirmed to be inhibited by serum from an SFTSV-infected patient (data not shown). – this brings up the point: did you incorporate positive and negative control sera into both of your SRIPs assays each time they were performed? This is not clear from the description in the Methods section.
It might be better to break figure 1 into 3 figures (maps, RLU, diagram of test principles) – I leave this to authors and editors. For Figure 1B, you should define what the error bar designates (standard error of the mean?) in the legend.
Line 161: “was greater than that in 2023.” You should indicate a p-value and the statistical method used for analysis.
Line 246: why would you think ELISA testing would require much more time than SRIPs?
You might mention in the discussion (with references) that JEV and SFTSV infections in swine are generally subclinical.
Another topic you might mention in discussion is whether there is any mosquito surveillance for JEV in Toyama or, although rare due to vaccination, are there data on human JEV or SFTSV incidence?
Author Response

(The authors gave the same response as above.)

Reviewer 3 Report
Comments and Suggestions for Authors
The objective of this manuscript entitled, “Prevalence of antibodies to Japanese encephalitis virus and severe fever with thrombocytopenia syndrome virus in wild boars captured across different locations in Toyama Prefecture” was to
determine the prevalence and distribution of antibodies against JEV and SFTSV in approximately 3,000 wild boar serum samples collected between 2019 and 2024. in Toyama. As a summary of the approach, methods and findings, the samples were originally collected and stored for classical swine fever testing. All samples were inactivated at 56°C for 30 minutes to prepare the test serum. A previously developed single-round infectious particles (SRIPs) assay system was employed to determine JEV and SFTSV neutralizing antibody positive samples. The results showed that the seroprevalence of a JEV and SFTSV neutralizing antibody increased annually from 2019 to 2024 and that JEV antibody-positive wild boars were distributed throughout Toyama Prefecture; whereas SFTSV antibody-positive wild boars were concentrated mainly in the northwestern region and along the western prefectural border. These findings suggested that JEV continued to pose an infection risk across the entire prefecture; while SFTSV has been actively spreading in the northwestern area during 2023–2024; raising concern over an increasing risk of human infection
Overall, the contents of this manuscript are readily understood, and the subject matter is suitable for publication consideration in the Journal Viruses. The approach need more detailed information, but the methods are scientifically sound and the results and discussion are original and contributes further to the understanding of the potential role of wild boars as source of human infection. In regard to the approach, as stated below in specific comments, suggest including a description of how the animals were captured, location, and how the age and sex was determined, and any other information collected on the animals, and how the samples were collected, processed and stored. As additional minor limitations, the manuscript could be improved by supplementing the abstract with more specific results for the antibody prevalence rates rather than just antibody positivity rates for both JEV and SFTSV increased annually from 2019 to 2024. As addressed below as a specific comment, was the SRIPs assay validated to determine sensitivity and specificity. Also, under specific comments, suggest supplementing the rationale and conclusion sections with what is known based on supporting evidence about wild boars as potential JEV and SFTSV amplifying host and any information available on the risk of these infected animals as a source of human JEV and/or SFTSV infection.
Specific comments
Lines 25-27 reads, “These findings suggest that JEV continue to pose an infection risk across the entire prefecture; while SFTSV has been actively spreading in the northwestern area during 2023–2024; raising concern over an increasing risk of human infection.”
Reviewer’s comment. Suggest replacing “suggest” with “suggested” and replace “continue” with “continued” to make past tense.
Lines 78-81 reads, “This study analyzed 3,059 serum samples obtained from wild boars captured in Toyama Prefecture between July 2019 and October 2024. These samples had been originally collected and stored for classical swine fever testing”.
Reviewer’s comment. Suggest to include a description of how the animals were captured, location, and how the age and sex was determined, and any other information collected on the animals, and how the samples were collected, processed and stored.
Lines 83-84 reads, “Consent for their use in this research was granted by the local governments and organizations responsible for capturing wild boars”.
Reviewer’s comment. Was the study conducted under an approved Institutional Animal Care and Use Protocol, suggest adding this to the manuscript as “consent” is not sufficient.
Lines 85 -105 Section 2.2 JEV-SRIPs and Lines 106-112 Section 2.3 SFTSV-SRIPs antibody assays are described as well as citing references regarding these assays in the Introduction section.
Reviewer’s comment. Were the assay validated to verified the sensitivity and specificity?
Lines 92-94 reads, “A 40 ul aliquot of the reaction mixture was inoculated onto Vero Osaka cells (obtained from the National Institute of Infectious Diseases, Japan)”.
Reviewer’s comment. Suggest adding the address of the source of the cells in Japan.
Lines 104-105 reads, “A regression line was constructed using the infection rates at each dilution, and the IC50 was determined, which was taken as the antibody titer”.
Reviewer’s comment. Suggest spelling-out IC50 and then abbreviating.
Lines 92-263 includes the Discussion section.
Lines 53-55 reads that pigs and wild boars act as amplifying hosts for JEV, and antibody positivity rate in these animals provides a useful indicator of the local presence and prevalence of the JEV. Also, lines 25-27 reads, these findings suggest that JEV continue to pose an infection risk across the entire prefecture; while SFTSV has been actively spreading in the northwestern area during 2023–2024; raising concern over an increasing risk of human, and lines 258-263 reads, by applying SRIPs-based methods to investigate antibody prevalence in a large number of wild animals and mapping the results, this study provided a clearer understanding of the risk of JEV and SFTSV transmission to humans in the study area. In Toyama Prefecture, JEV remains a public health risk, while SFTSV is currently concentrated in the northwestern region but appears to be spreading, indicating a rising risk of SFTSV infection.
Reviewer’s comment. This information lacks detailed supporting evidence of the role of JEV and/or SFTSV infected wild boars as a risk of human infection/disease. Suggest including what is known based on supporting evidence about wild boars as potential virus amplifying host and any information available of the risk of these infected animals as a source of human JEV and/or SFTSV infection.
Author Response

(The authors gave the same response as above.)

Round 2
Reviewer 1 Report
Comments and Suggestions for Authors
I would like to thank the authors for the explanations and for modifying their paper according to most of my indications.
The introduction and methods section are fine in the current version. The results section is still requiring some revision. Indeed, the current version now begins with table 1, but the supporting text
has been retroceded to line 99 (methods). So, now, there is no text for the results section until after table 1 and figure 1. So, since these two are supported by text shown in the methods, they should maybe
also be moved to that section. I still have some doubts about figure 1B's usefulness. As a minor point, please consider that in fig 3 there is no dashed line, in contrast with the statement in the caption.
The discussion is acceptable.
I invite the authors to try to perform the suggested small modifications to ensure paper acceptation
Author Response
Thank you for pointing out. Please see the attachment.
